# Rapid Benzylation of Wood Powder without Heating

**DOI:** 10.3390/polym13071118

**Published:** 2021-04-01

**Authors:** Mitsuru Abe, Masako Seki, Tsunehisa Miki, Masakazu Nishida

**Affiliations:** Multi-Materials Research Institute, National Institute of Advanced Industrial Science and Technology (AIST), 2266-98 Shimoshidami, Moriyamaku, Nagoya 463-8560, Japan; m-seki@aist.go.jp (M.S.); tsune-miki@aist.go.jp (T.M.); m-nishida@aist.go.jp (M.N.)

**Keywords:** hydroxide, benzylation, wood, rapid reaction, thermoplasticity

## Abstract

Converting wood waste into thermoplastic materials is an attractive means of increasing its utilization. A promising method for imparting thermoplasticity to wood is chemical modification, in which the hydroxyl groups in wood are substituted with benzyl groups. In the common method, wood powder is first treated with a highly concentrated aqueous NaOH solution, and then reacted with a benzylation reagent by heating for a long time under stirring. In this study, a 50% aqueous tetra-*n*-butylphosphonium hydroxide solution was used for the pretreatment of wood powder. This modified alkaline treatment enhanced the efficiency of the subsequent benzylation reaction, which could be conducted without heating over a shorter time. The effects of various conditions on the efficiency of the benzylation reaction were evaluated. Both the alkali pretreatment and the subsequent benzylation required only ~5–10 min of stirring without heating to obtain benzylated wood with a similar degree of benzylation as that achieved by the common method. The chemical structure of the benzylated wood powder was characterized by Fourier-transform infrared and solid-state NMR spectroscopies, and its thermal softening characteristics were evaluated by thermomechanical analysis. Finally, a translucent film could be obtained by hot-pressing the benzylated wood powder.

## 1. Introduction

Wood is an abundant renewable resource on Earth and is known to be a carbon cycle material. Therefore, further utilization of wood is essential to build a sustainable society. However, as wood cannot be melted by heating, it cannot be plastically molded, which limits its use in various applications. One of the strategies for the plastic molding of wood involves mixing it with a thermoplastic resin. Therefore, the development of wood-plastic composites by kneading powdered wood powder and resin is being actively researched [1,2,3,4,5]. Further, the chemical modification of wood to render it thermoplastic is also being explored as a method to limit the amount of petroleum-derived resin used in the composites. Esterification is a commonly used method for the thermoplasticization of wood [6,7,8,9,10,11]. On the other hand, as ether groups are generally more stable than the ester groups, etherification may be preferred as an alternative route to functionalize wood to render it thermoplastic. In particular, benzylation is expected to be a promising chemical modification reaction that can impart wood with thermoplasticity and water resistance.

The thermoplasticization of wood by benzylation has been studied since the 1980s. In 1983, Norimoto et al. succeeded in benzylating poplar powder to impart thermoplasticity [12]. They first soaked wood powder in a 40% aqueous sodium hydroxide (NaOH) solution. Subsequently, they squeezed the sample and reacted it with benzyl chloride (BnCl) at 110 °C for 3–6 h under stirring to obtain benzylated wood. Hon et al. reported that pine could be also subjected to a similar benzylation treatment to render it thermoplastic [13]. They treated degreased pine powder (40–60 mesh) with 15–40% aqueous NaOH solution, and then reacted it with BnCl at a high temperature of 110 °C under stirring. They reported that a high alkali concentration, high reaction temperature, and long heating time are preferable for obtaining highly benzylated wood. Currently, many researchers follow the Hon’s method for benzylation of wood.

Further, several studies have also been conducted on the benzylation of lignocelluloses, including wood. Pereira et al. [14] successfully obtained a benzylated sugar cane bagasse pulp using a procedure similar to that reported by Hon et al. They mixed 2 g of a pulp sample containing 9% lignin with 35 mL of a 40% aqueous NaOH solution. The mixture was subsequently heated to 110 °C, 15 mL of BnCl was added, and then stirred for several hours. They reported a decrease in the weight of the benzylated pulp until the first 3 h, followed by a rapid increase in weight. The observed weight changes were attributed to the combination of weight loss due to the degradation of some components and the weight gain due to benzylation. Chen et al. performed a similar benzylation treatment on kenaf fibers [15] and noted that heating and stirring at 110 °C for at least 3 h was required to achieve a certain degree of benzylation. They also succeeded in molding the benzylated kenaf fiber into a sheet by hot-pressing and investigated the mechanical strength of the sheet. Üner et al. benzylated sawdust using a similar procedure to obtain a thermoplastic material. They evaluated the effects of the type and concentration of the alkali hydroxide used in the pretreatment. They reported that no significant differences were observed with the use of NaOH, LiOH, and KOH. However, guanidine was less effective than the three alkaline reagents tested. Furthermore, they reported that a higher concentration of the alkaline reagent in the range of 15–35% is preferable [16]. Although benzylation is a promising method for imparting thermoplasticity to lignocellulose, including wood, it requires prolonged heating and stirring, as described above; therefore, the benzylation method is not practically applicable.

Our group developed a method for rapidly etherifying the hydroxyl groups of cellulose without heating [17,18]. In our studies, we used an aqueous hydroxide solution with a tetra-*n*-butylphosphonium cation ([(*n*-Bu)_4_P]), which strongly interacts with the hydroxyl proton of cellulose, as the solvent for the etherification reaction. When cellulose was dissolved in a 50% aqueous [(*n*-Bu)_4_P]OH solution and then treated with benzyl bromide (BnBr), ~80% of the hydroxyl groups in cellulose were substituted with benzyl groups in ~10 min at 25 °C. The benzylated cellulose precipitated as a white powder in the reaction solution. We hypothesized that if this technique could be applied to wood, it could be easily and rapidly benzylated to render it thermoplastic. However, because it is difficult to completely dissolve wood in a 50% aqueous [(*n*-Bu)_4_P]OH solution [19], the etherification of wood is expected to be a biphasic solid-liquid reaction. In addition, as wood contains various components, such as hemicellulose and lignin, other than cellulose, it cannot be predicted whether the etherification reaction proceeds as in the case of pure cellulose.

In this study, we aimed to develop a new method for the benzylation of wood powder, without prolonged heating and stirring, using a 50% aqueous [(*n*-Bu)_4_P]OH solution as the pretreatment reagent. Further, by evaluating the effects of various reaction conditions on the degree of benzylation and then optimizing the treatment conditions, we achieved efficient benzylation of wood powder at a lower temperature and for a significantly shorter duration. We also succeeded in fabricating a film of the benzylated wood powder obtained by this method.

## 2. Materials and Methods

### 2.1. Original Materials

Japanese cedar (*Cryptomeria japonica*) was used as the raw material for benzylation. Air-dried cedar was ground into a powder (50–100 mesh) using a Wiley mill. Methanol, NaOH, BnCl, and BnBr were purchased from FUJIFILM Wako Pure Chemical Corp. (Osaka, Japan) and used as received. A 40% aqueous [(*n*-Bu)_4_P]OH solution was purchased from the same company and concentrated to 50% at 35 °C under vacuum, before use.

### 2.2. Preparation of Wood Samples

To degrease the wood powder, Soxhlet extraction was conducted using methanol for 24 h and then hot water for 24 h. Then, the sample was washed with distilled water and the obtained degreased wood was dried at 35 °C under vacuum for 24 h (entry 0 in Table 1).

The benzylation of wood by the common method was conducted as follows: Degreased wood powder (0.5 g) was mixed with 4 mL of a 40% aqueous NaOH solution, and the mixture was stirred at 25 °C for 5 min. Then, 4 mL of BnCl was added to the mixture and stirred at 110 °C for 2 h. The obtained product was purified by Soxhlet extraction using methanol for 24 h and then hot water for 24 h, followed by washing with distilled water. Finally, the obtained benzylated wood was dried at 35 °C under vacuum for 24 h (Table 1, entry 1).

The typical procedure for the novel benzylation method is as follows. Degreased wood powder (0.5 g) was mixed with 4 mL of a 50% aqueous [(*n*-Bu)_4_P]OH solution, and the mixture was stirred at 25 °C for 5 min. Subsequently, 4 mL of BnBr was added to the mixture and stirred at 25 °C for 1 h. The washing and drying procedures were the same as those described above (Table 1, entry 6).

All the syntheses were conducted multiple times, and the experimental error was determined.

### 2.3. Fourier-Transform Infrared (FT-IR) Spectroscopy

FT-IR spectra were recorded on a Nicolet 6700 spectrometer (Thermo Scientific Inc., Waltham, MA, USA) at 4 cm^−1^ resolution in the standard attenuated total reflectance mode; 32 scans were made in the range of 4000–500 cm^−1^. The spectra were obtained three or more times for each sample, and no noticeable differences were observed in the results.

### 2.4. NMR Spectroscopy

Solid-state ^13^C NMR spectra were recorded on a Varian 400 NMR system spectrometer (Palo Alto, CA) operated at 100.56 MHz and equipped with a Varian 4 mm double-resonance T3 solid probe for the ^13^C nuclei. The spectra were collected with a 40 ms acquisition period over a 30.7 kHz spectral width over the temperature range of 20 to 22 °C. Proton decoupling was performed with an 86 kHz ^1^H decoupling radio frequency with a small phase incremental alteration (SPINAL) decoupling pulse sequence. Cross-polarization and magic angle spinning (CP-MAS) NMR studies were conducted with a 5.0 s recycle delay in 1024 transients using a ramped-amplitude pulse sequence with a 2 ms contact time and a 2.6 μs π/2 pulse for the ^1^H nuclei. The amplitude of the ^1^H nuclei was ramped down linearly from 92.6% of its final value during the cross-polarization contact time. Pulse saturation transfer and magic angle spinning (PST-MAS) NMR measurements were conducted in 2048 transients with a 4.8 μs π/2 pulse for the ^13^C nuclei with a 5 s recycle delay after the saturation of the ^1^H nuclei with five consecutive 2.6 μs pulses and a 27.5 μs delay.

### 2.5. Thermomechanical Analysis (TMA)

TMA thermograms were recorded using a TMA SS6100 instrument (Hitachi High-Tech Science Corp., Tokyo, Japan). The dried wood sample (0.01 g) was placed in an open aluminum pan (*ϕ* = 5 mm), and the wood powder was pressed firmly using a cylindrical aluminum rod (*ϕ* = 4.9 mm). Thereafter, it was covered with a thin flat aluminum plate (*ϕ* = 4.9 mm) and used for TMA analysis. The temperature was increased in the range of 30–250 °C at 1 °C/min rate. The temperature was ramped up while applying a load of 300 mN using a quartz rod, and the descent of the rod was measured.

### 2.6. Synthesis of A Translucent Film

The benzylated samples were pressed between two pieces of flat dies using a small heat press machine with cooling function (HC300-15, AS ONE Corp., Osaka, Japan). The hot press machine and dies were heated to 160 °C before use. After 10 min of temperature equilibration, 0.04 g of the dried wood sample was set on the press surface of the flat die. Then, it was sandwiched between the flat dies and pressed with a force of 10 t for 5 min. After that, the dies were cooled to approximately 25–35 °C, and pressed wood samples were collected.

## 3. Results and Discussion

### 3.1. Benzylation of Wood

First, with reference to the report by Hon et al., the wood powder was benzylated under the commonly employed treatment conditions [13] (entry 1 in Table 1). The weight percentage gain (WPG) of the benzylated wood powder was ~20% of that of the precursor. Figure 1 shows the FT-IR spectra of the precursor (degreased wood powder) and benzylated wood samples. The peak intensity of the hydroxyl groups (3000–3600 cm^−1^) decreased significantly after the benzylation treatment. On the other hand, IR peaks attributable to benzyl groups appeared at 3020–3080, 1592–1612, 1454–1496, 736, and 696 cm^−1^ (Figure 1, blue line) [12]. These changes indicate that the hydroxyl groups were replaced with benzyl groups. The IR spectrum of the precursor (entry 0) shows an IR peak at 1730 cm^−1^, which is attributed to the acetyl and carboxyl groups of hemicelluloses and aldehyde groups or aliphatic ketones in lignin. In addition, an IR peak at 1660 cm^−1^, which is attributed to the conjugated carbonyl groups in lignin, was also detected (entry 0) [20]. The intensities of these peaks decreased after the benzylation reaction (entry 1) because of the alkaline treatment. The acetyl groups were cleaved from hemicellulose during the treatment. In addition, the carbonyl groups in lignin were probably oxidized to carboxyl groups and then to carboxylates. Because of this oxidative decomposition, a certain amount of lignin dissolved in the alkaline solvent. As suggested by Hon et al., the WPG might include the effects of several reactions [13]. In order to accurately estimate the reactivity of the benzylation step, the relative peak intensities in the FT-IR spectra were analyzed in this study. The decrease in the rate of the hydroxyl group-derived peaks and the increase in the rate of the aromatic ring-derived peaks were calculated. The decrease rate of the OH-derived peak was calculated from the ratio of the peak intensity derived from the OH stretching vibrations appearing at 3000–3600 cm^−1^ to the peak intensity derived from CH stretching vibrations appearing at 2800–3000 cm^−1^ (OH/CH intensity ratio, Figure 2). The increase rate of the peaks derived from the benzene rings was calculated from the ratio of the peak intensity derived from the mono-substituted benzene rings that appeared at 680–714 cm^−1^ to the peak intensity derived from the CH stretching vibrations (Bn/CH intensity ratio, Figure 2). The smaller the OH/CH intensity ratio and the larger the Bn/CH intensity ratio, the greater the number of hydroxyl groups replaced with benzyl groups. The results are presented in the right column of Table 1. The calculated OH/CH intensity ratio of the degreased wood powder (entry 0) is 3.4, and the Bn/CH intensity ratio is 0.1, while the corresponding values for the benzylated wood (entry 1) are 0.8 and 4.9, respectively.

Next, the wood powder was benzylated under the same treatment conditions as in entry 1 using a 50% aqueous [(*n*-Bu)_4_P]OH solution as the pretreatment solution (entry 2 in Table 1). This sample yielded an OH/CH intensity ratio of 2.1 and a Bn/CH intensity ratio of 3.0 (entry 2). The OH/CH intensity ratio is smaller, and the Bn/CH intensity ratio is larger than the corresponding ratios of the precursor (entry 0), implying that benzylation occurred successfully in the case of entry 2 (Figure 1, green). However, more hydroxyl groups were substituted with benzyl groups in the case of entry 1 than in the case of entry 2. In summary, in the case of benzylation under heating and long-term stirring, as in the common method, a 40% aqueous NaOH solution yields better results than the 50% aqueous [(*n*-Bu)_4_P]OH solution as a pretreatment reagent.

Further, wood benzylation was also examined at a lower temperature and a shorter reaction time (entries 3 and 4 in Table 1). When a 40% NaOH solution was used for the pretreatment, benzylation hardly occurred (entry 3). On the other hand, when a 50% [(*n*-Bu)_4_P]OH solution was used, the OH/CH and Bn/CH intensity ratios were 1.3 and 3.3, respectively (entry 4). Compared to the case of entry 2, that is, treatment at 110 °C for 2 h, a greater number of hydroxyl groups were substituted with benzyl groups. Thus, when a 50% [(*n*-Bu)_4_P]OH solution was used for the pretreatment, the wood powder was successfully benzylated at room temperature, although it contained hemicellulose and lignin in addition to cellulose.

Next, the degree of benzylation of wood at lower temperatures and shorter times were evaluated using BnBr as the benzylating reagent, because BnBr is expected to react more efficiently than BnCl. First, the benzylation reaction was conducted under conditions similar to those in entry 1 (see entry 5 in Table 1). The OH/CH and Bn/CH intensity ratios of the obtained wood powder were 2.9 and 0.3, respectively, implying that the degree of substitution was very low. Subsequently, a similar experiment was conducted using a 50% [(*n*-Bu)_4_P]OH solution for pretreatment (entry 6 in Table 1). In this case, the OH/CH and Bn/CH intensity ratios of the obtained benzylated wood were 0.7 and 5.3, respectively. This implies that the benzylation of wood occurred to a significant extent (Figure 1, red). Finally, using the 50% [(*n*-Bu)_4_P]OH solution as the pretreatment reagent, benzylated wood powder with a degree of substitution comparable to or higher than that achieved with the common method (entry 1) was obtained at room temperature, without heating.

### 3.2. Effects of Reaction Conditions

The effects of various reaction conditions on the reactivity in the benzylation of wood powder using the 50% aqueous [(*n*-Bu)_4_P]OH solution were evaluated by varying the amount of BnBr, reaction temperature, and reaction time.

#### 3.2.1. In-Feed Amount of BnBr

By maintaining the basic treatment conditions the same as for entry 6, the amount of BnBr was changed between 0.5 and 8 mL for the benzylation of wood powder. The corresponding results are shown in Figure 3. When the amount of BnBr was in the range of 0.5–2 mL, the OH/CH intensity ratio (Figure 3, blue circle) and Bn/CH intensity ratio (Figure 3, orange triangle) changed significantly depending on the BnBr amount. On the other hand, when the amount of BnBr was in the range of 2–8 mL, neither the OH/CH intensity ratio nor the Bn/CH intensity ratio changed; they were 0.7–0.8 and 5.1–5.2, respectively. These results suggested that if the amount of BnBr is 2 mL or more, the benzylation proceeds to the same extent as that of the common method (entry 1 in Table 1), even at 25 °C.

#### 3.2.2. Reaction Temperature

Next, the effect of the reaction temperature on the benzylation treatment was evaluated. Based on the above results, the amount of BnBr was fixed at 4 mL, which is slightly higher than the minimum amount required. Other conditions were maintained the same as those in entry 6. Figure 4 shows the results obtained by varying the reaction temperature of the benzylation treatment between 4 and 100 °C. When the reaction was performed in the range of 4–50 °C, both the OH/CH and Bn/CH intensity ratios were almost constant (Figure 4), indicating that the effect of the reaction temperature was not significant in this range. On the other hand, at 75 °C or higher, the OH/CH intensity ratio increased, and the Bn/CH intensity ratio decreased as the temperature increased, implying that the degree of substitution was low. A similar result was observed with the use of BnCl as the benzylation reagent (comparison of entries 2 and 4). That is, when a 50% [(*n*-Bu)_4_P]OH solution was used for the pretreatment, the benzylation of wood proceeded easily under mild conditions, that is, at room temperature. In a previous study, a similar tendency was observed in the case of the etherification of dissolved cellulose [17]. Although the reason for this tendency is not yet clear, it is possible that some of the [(*n*-Bu)_4_P]OH was thermally decomposed at higher temperatures. Another possibility of low degree of substitution is the inactivation of BnBr by [(*n*-Bu)_4_P]OH. BnBr reacts with the OH anion of the strong basic solvent to form BnOH, which does not react with wood. This inactivation of BnBr might have been promoted at higher temperature.

#### 3.2.3. Reaction Time

Next, the effect of the processing time was evaluated. First, the pretreatment time with the alkaline aqueous solution was varied between 5 and 60 min. The other conditions were set the same as those in entry 6. The OH/CH intensity ratio was approximately 0.7, and the Bn/CH intensity ratio was approximately 5.3, even with a very short pretreatment time of 5 min (Figure 5). In addition, the OH/CH and Bn/CH intensity ratios hardly changed even when the pretreatment time was prolonged further. Thus, the pretreatment time had almost no effect on the extent of benzylation in the examined range.

The effect of the BnBr treatment time on the degree of benzylation was also evaluated. From the results of the previous section, the pretreatment time with the alkaline aqueous solution was set to 10 min, and the other reaction conditions were set in the same manner as those in entry 6. The wood powder was benzylated by varying the stirring time between 5 and 60 min after the addition of BnBr. As shown in Figure 6, benzylated wood powder with an OH/CH intensity ratio of ~0.7, and a Bn/CH intensity ratio of ~4.9, was obtained after a short time of only 5 min. Further, the OH/CH intensity ratio did not change with a further increase in the reaction time. Meanwhile, the Bn/CH intensity ratio increased slightly with the prolongation of the reaction time; the value was ~5.4 after 1 h of treatment. However, the difference was very small, indicating that the benzylation reaction was completed during the first few minutes. Therefore, the [(*n*-Bu)_4_P]OH pretreatment solution facilitated the benzylation of wood with a conversion rate similar to that of the common method (entry 1) in a very short duration, even in the absence of heating.

### 3.3. Characterization of the Benzylated Wood

Next, the chemical and physical properties, including thermoplasticity, of the obtained benzylated wood powder were characterized.

#### 3.3.1. Solid-State NMR Measurement

Figure 7 shows the ^13^C CP-MAS NMR spectra of the degreased wood powder (entry 0) and benzylated wood powders (entries 1 and 6). Signals of biomass constituents in the degreased wood were assigned with reference to our previous report [21]. The signals of the cellulose units in the degreased wood were observed at 105 ppm (C1), 89 ppm (crystalline C4), 84 ppm (amorphous C4), 75 and 72 ppm (C2, 3, 5), 65 ppm (crystalline C6), and 63 ppm (amorphous C6). The signals of hemicellulose overlapped with those of cellulose at 103 ppm (C1), 75 ppm (C2–4), and 64 ppm (C5). The lignin signals appeared separated from those of the carbohydrates at 160–110 ppm (aromatic) and 56 ppm (-OCH_3_), although the former signals appeared as broad signals. After benzylation, the NMR signals derived from 2,3,4,5-C_6_H_5_ and 1-C_6_H_5_ of the benzyl groups appeared at 128 and 139 ppm, respectively. An NMR signal derived from the CH_2_ of the benzyl groups overlapped with the signal derived from the intracycle carbons (cellulose C2, 3, 5, and hemicellulose C2–4) in the carbohydrates.

Benzylation changed the signals of the biomass constituents. According to the C4 signals of cellulose, benzylation decreased the crystallinity of the cellulose unit. In this case, the cellulose crystallinity for the case of entry 1 was lower than that for the case of entry 6. The benzylation of wood also decreased the overlapped signals of cellulose and hemicellulose (105 ppm and 63–65 ppm) as compared to the signals of cellulose only (84–89 ppm), indicating that benzylation significantly decreased the hemicellulose content. Although benzylation decreased the aromatic signal of lignin (150 ppm), the decrease rate of lignin was lower than the decrease rate of hemicellulose. The decrease rate of lignin for entry 6 was higher than that for entry 1. Therefore, benzylation with both the NaOH and [(*n*-Bu)_4_P]OH solutions led to the decomposition of the hemicellulose units in the original wood. Comparison of the reactivity of biomass constituents in the original wood in these two methods indicated that the NaOH solution decomposed the crystalline form of cellulose to a greater extent, while the [(*n*-Bu)_4_P]OH solution decomposed the lignin unit to a greater extent.

The PST-MAS method enhances the signals of flexible components with high mobility near hydrogen atoms, such as alkyl or alkoxy side chains [22]. Figure 8 shows the ^13^C PST-MAS NMR spectra of degreased wood powder (entry 0) and benzylated wood powders (entries 1 and 6). According to the measurement on the empty rotor, the largest signal at 113 ppm was assigned as the background signal originating from the sample rotor and probe. The ^13^C PST-MAS NMR spectra of degreased wood shows three small signals at 75 ppm (cellulose C2, 3, 5, and hemicellulose C2–4), 63 ppm (cellulose C6 and hemicellulose C5), and 56 ppm (lignin -OCH_3_) because the intracycle carbons have lower molecular mobility than the terminal carbons. Furthermore, carbons bridging the sugar ring (C1 and C4) could not be detected in the ^13^C PST-MAS NMR spectra because of rigid glycosidic bonds.

After benzylation, the intensity of the ^13^C PST-MAS signals derived from cellulose C6 and hemicellulose C5 (63 ppm) decreased. On the other hand, the signal intensity derived from the benzyl ring carbons (2,3,4,5-C_6_H_5_) near 128 ppm was larger than that of the 1-C_6_H_5_ carbon (Figure 8, entries 1 and 6). The CH_2_ group of the benzyl group has relatively high mobility, although the overlapped intracycle carbons of the carbohydrate have low mobility, resulting in an increase in the ^13^C PST-MAS signals at 75 ppm. These results suggest that the benzyl groups introduced into the original wood have relatively high motility, especially the aromatic ring. Comparison of the results for entry 1 and 6 indicated no differences in the PST-MAS data for the two samples, including the signal intensity of the carbohydrates, although the CP-MAS data indicated differences in the crystalline cellulose and lignin contents. In other words, there was no significant difference in the molecular mobility of the benzylated wood prepared by the common method using NaOH (entry 1) and that prepared in a short time using [(*n*-Bu)_4_P]OH, without heating (entry 6).

#### 3.3.2. TMA

Next, the thermodynamic properties of the degreased and benzylated woods were evaluated by TMA. Figure 9 shows the displacement (μm) of the quartz rod owing to the softening of each sample. In the case of degreased wood (entry 0), there were no inflection points, and the rod sank gradually as the temperature increased. The benzylated wood obtained using the NaOH solution (entry 1) showed an inflection point at 95 °C, after which the displacement per unit temperature rise increased. Norimoto et al. reported that the inflection point corresponds to a change in the benzylated wood from a glass state to a rubber flow state [12]. Meanwhile, the benzylated wood obtained using the [(*n*-Bu)_4_P]OH solution (entry 6) also showed an inflection point at 105 °C; however, the displacement of the quartz rod was smaller than that observed with the case of entry 1. The smaller displacement was caused by the higher crystallinity of the cellulose. Nevertheless, the TMA results suggest that the benzylated woods obtained using both the NaOH and [(*n*-Bu)_4_P]OH solutions have a softening point, implying that they exhibit thermoplasticity. Thus, wood was rendered thermoplastic even with the low-temperature benzylation using the [(*n*-Bu)_4_P]OH solution for pretreatment.

#### 3.3.3. Fabrication of a Translucent Film

To confirm that the benzylated wood is thermoplastic, we attempted to obtain a film by hot pressing. When the degreased wood (entry 0) was pressed between two dies, a mass of wood powder was obtained (Figure 10, left). As the wood powder was not fused, it crumbled easily when pinched with the fingers. On the other hand, the benzylated wood powder obtained using the [(*n*-Bu)_4_P]OH pretreatment solution (entry 6) melted between the flat dies heated at 160 °C and fused to form a translucent film (Figure 10, right). This result clearly suggests that low-temperature benzylation provides a wood powder with thermoplasticity, which can be molded.

As demonstrated above, the benzylation method using the phosphonium cation developed in this study was able to impart the wood powder with thermoplasticity. Benzylation with the phosphonium cation proceeded under mild conditions (room temperature and a short reaction time), whereas the previous common method required a higher reaction temperature and a long stirring time. Rapid benzylation under mild conditions will be applicable to materials that are unstable at high reaction temperatures. Next, we plan to study the forming and fabrication processes of materials with thermoplasticity using this rapid and mild benzylation method.

## 4. Conclusions

Efficient benzylation of wood powder was accomplished without heating and in a short time with the use of a 50% aqueous [(*n*-Bu)_4_P]OH solution as a pretreatment solution. The degree of substitution was affected by the amount of the benzylation reagent used; the ratio of wood powder to BnBr is preferably 1:4 in wt%. The optimal reaction temperature for the benzylation was found to be 4–50 °C, and the reaction efficiency decreased at a temperature of 75 °C or higher. Regarding the treatment time, both the alkali treatment and the benzylation treatment were completed in approximately 5–10 min. The wood sample was benzylated to almost the same degree as that of the sample benzylated by the common method. According to the solid-state NMR studies, benzylation using the [(*n*-Bu)_4_P]OH pretreatment solution maintained the crystallinity of the cellulose and slightly decomposed the lignin unit, as compared with the common method. The obtained benzylated wood powder could be heat-softened at approximately 90–100 °C to obtain a translucent film by hot pressing. This benzylation method renders thermoplasticity to wood in a short time without heating, therefore it is expected to contribute to promoting the use of wood as an industrial material.

## Figures and Tables

**Figure 1 polymers-13-01118-f001:**
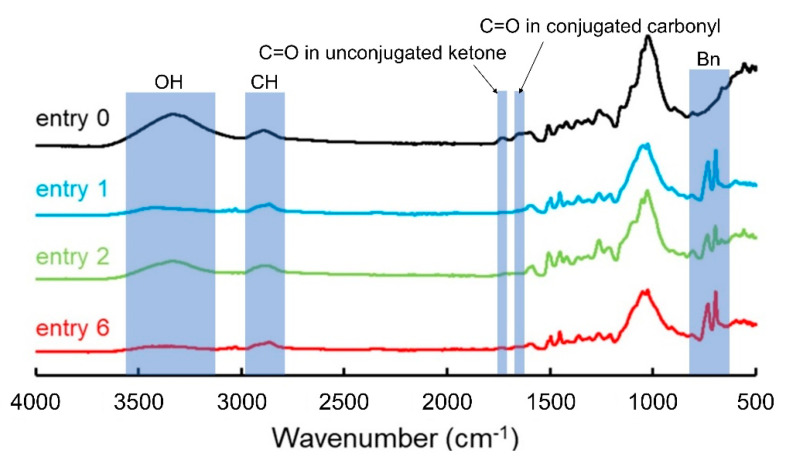
FT-IR spectra of degreased wood (entry 0, black) and benzylated woods (entry 1, blue; entry 2, green; and entry 6, red).

**Figure 2 polymers-13-01118-f002:**
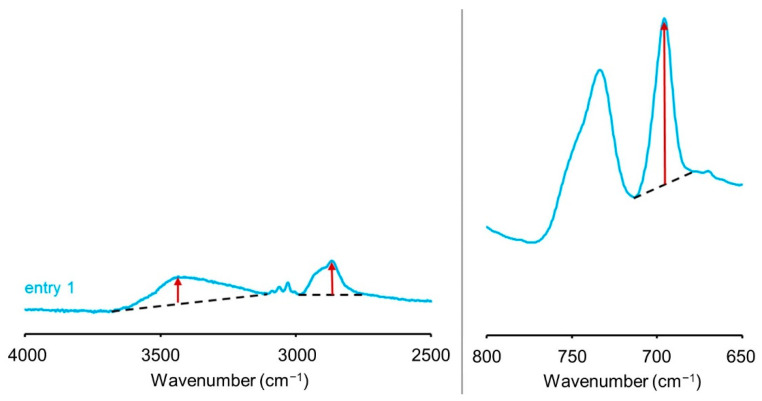
Intensities of peaks at approximately 3400, 2870, and 696 cm^−1^ in the IR spectra of benzylated wood (entry 1).

**Figure 3 polymers-13-01118-f003:**
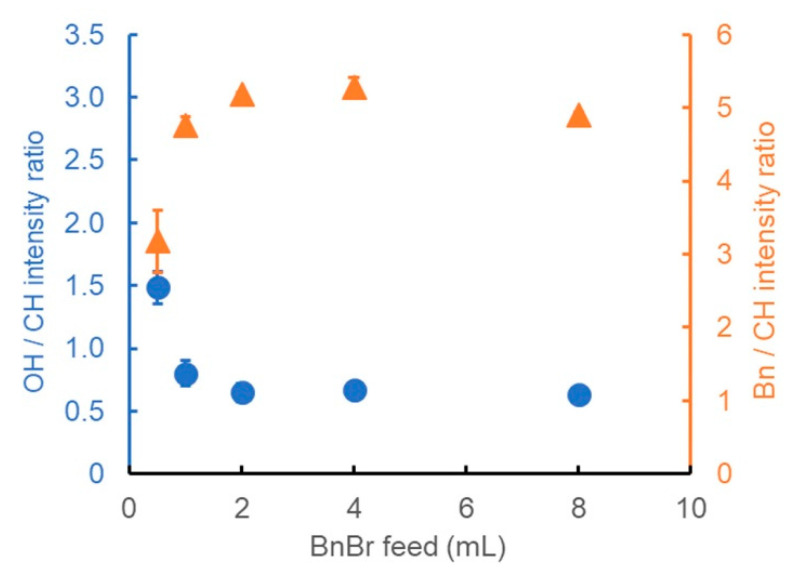
Ratio of the IR peak intensity derived from the OH groups (blue circles) or benzene rings (orange triangles) to that of CH stretching vibrations, showing the dependence of the degree of benzylation on the BnBr amount

**Figure 4 polymers-13-01118-f004:**
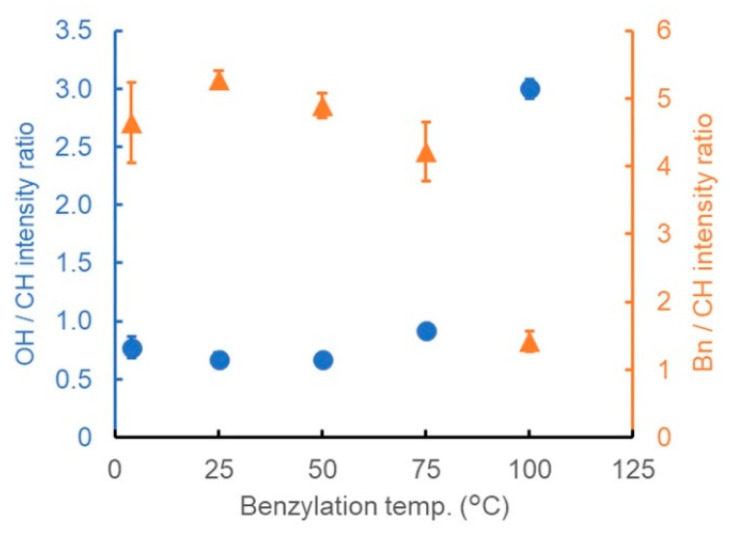
Ratio of the peak intensity derived from the OH groups (blue circles) or benzene rings (orange triangles) to that of CH stretching vibrations, showing the dependence of the degree of benzylation on the reaction temperature.

**Figure 5 polymers-13-01118-f005:**
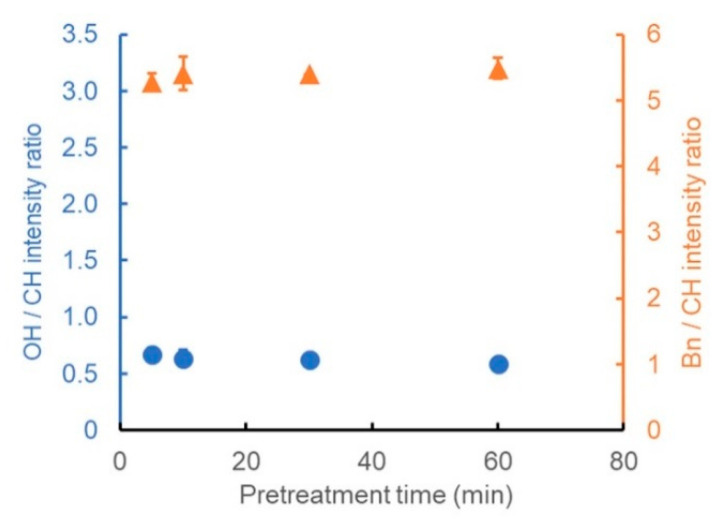
Ratio of the peak intensity derived from the OH groups (blue circles) or benzene rings (orange triangles) to that of CH stretching vibrations, showing the dependence of the degree of benzylation on the pretreatment time.

**Figure 6 polymers-13-01118-f006:**
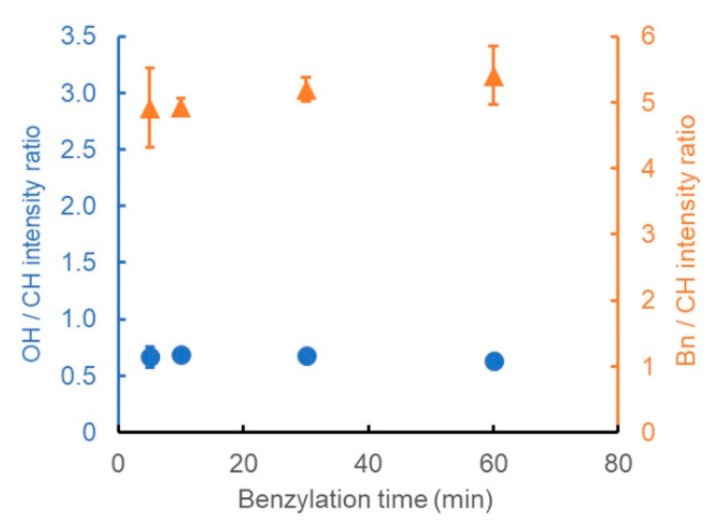
Ratio of the peak intensity derived from the OH groups (blue circles) or benzene rings (orange triangles) to that of CH stretching vibrations, showing the dependence of the degree of benzylation on the benzylation time.

**Figure 7 polymers-13-01118-f007:**
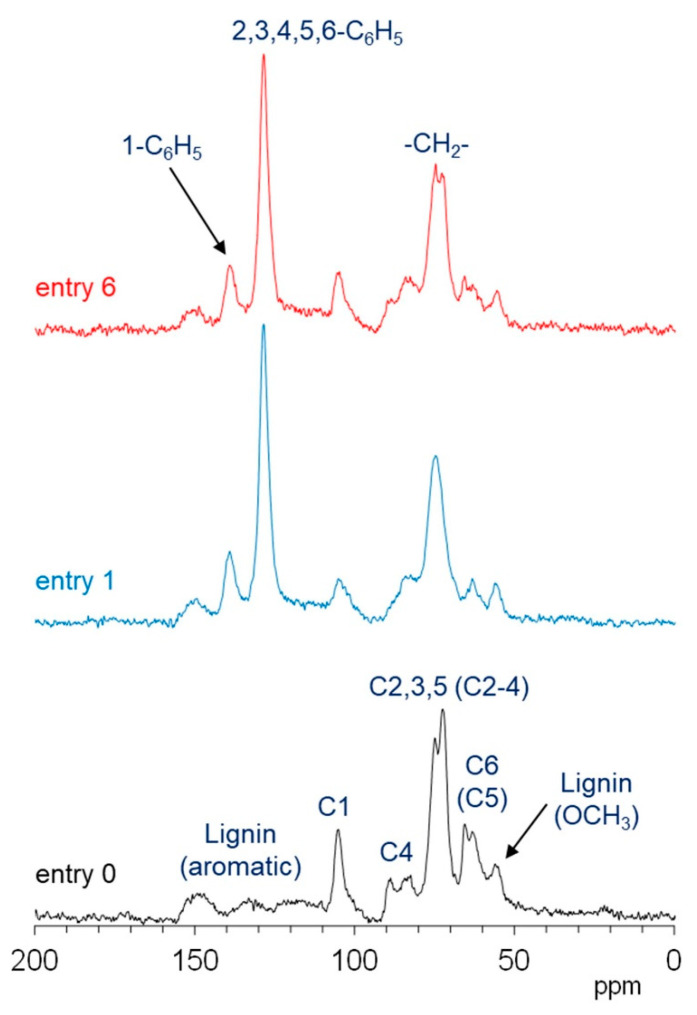
^13^C CP-MAS NMR spectra of degreased wood powder (entry 0, black) and benzylated wood powders (entry 1, blue; and entry 6, red).

**Figure 8 polymers-13-01118-f008:**
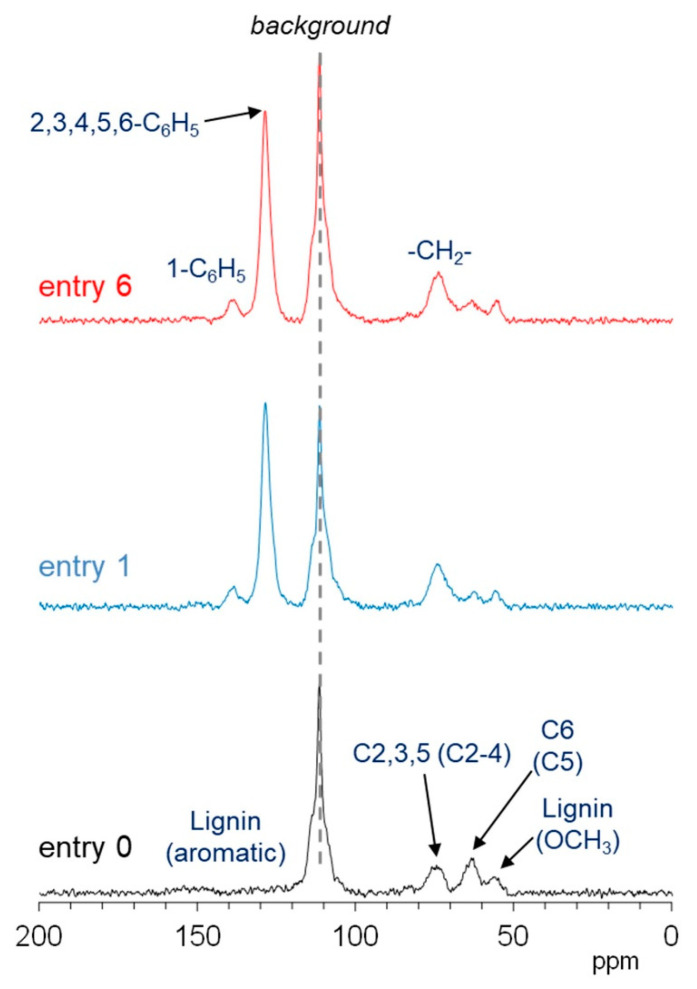
^13^C PST-MAS NMR spectra of degreased wood powder (entry 0, black) and benzylated wood powders (entry 1, blue; and entry 6, red).

**Figure 9 polymers-13-01118-f009:**
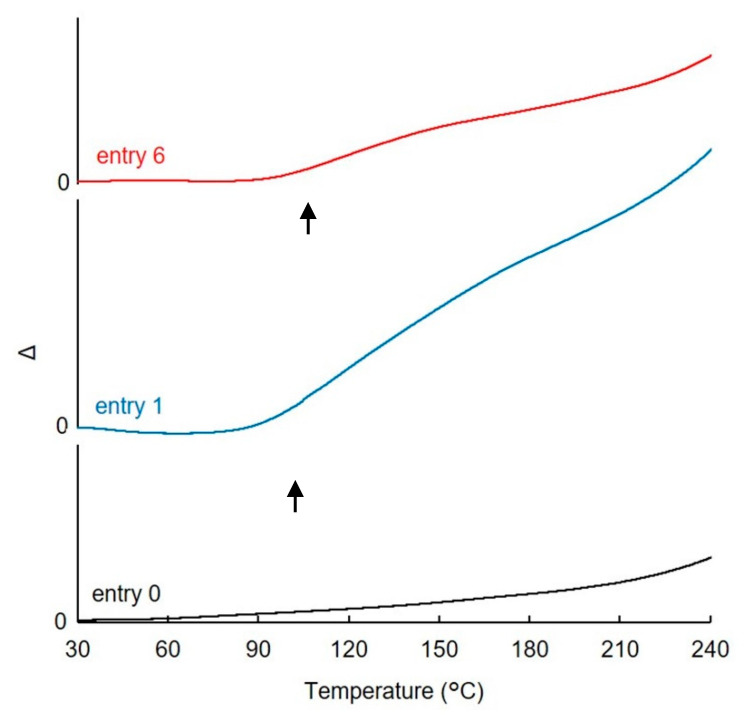
TMA thermograms of degreased wood (entry 0) and benzylated woods (entry 1 and 6). Arrows indicate the thermal softening points.

**Figure 10 polymers-13-01118-f010:**
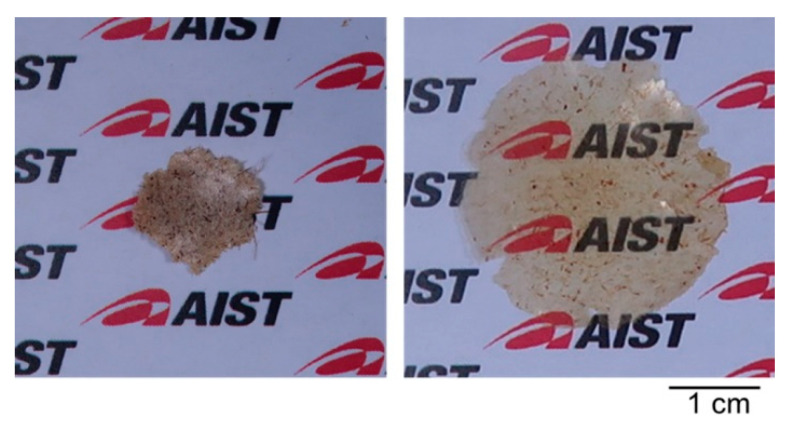
Photographs of a mass obtained by pressing degreased wood powder (**left**, entry 0) and a translucent film obtained by pressing wood powder benzylated at a low temperature using an [(*n*-Bu)_4_P]OH solution (**right**, entry 6).

**Table 1 polymers-13-01118-t001:** Treatment conditions of benzylation and the intensity ratio of the FT-IR peaks.

Sample	Alkaline Treatment			Benzylation Treatment			Intensity Ratio of FT-IR Peaks	
	Solvent	Temp (°C)	Time (min)	Reagent	Temp (°C)	Time (min)	OH/CH	Bn/CH
entry ^a^ 0	-	-	-	-	-	-	3.4	0.1
entry 1	40% NaOH	25	5	BnCl	110	120	0.8	4.9
entry 2	50% [(*n*-Bu)_4_P]OH	25	5	BnCl	110	120	2.1	3.0
entry 3	40% NaOH	25	5	BnCl	25	60	3.3	0.2
entry 4	50% [(*n*-Bu)_4_P]OH	25	5	BnCl	25	60	1.3	3.3
entry 5	40% NaOH	25	5	BnBr	25	60	2.9	0.3
entry 6	50% [(*n*-Bu)_4_P]OH	25	5	BnBr	25	60	0.7	5.3

^a^ Only degreasing treatment was conducted for the wood sample.

## Data Availability

The data presented in this study are available on request from the corresponding author.

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
