# Peer review of "Rapid Benzylation of Wood Powder without Heating"

_polymers, 2021, doi:10.3390/polym13071118_

Round 1

Reviewer 1 Report

Dear Editor

In the current study, the researchers have proposed a new approach for the etherification of wood to impact thermoplasticity. The approach is quite interesting and will save both energy and resource on an industrial scale. The study is well presented. Especially the authors have successfully given all the background info and hypothesis in the introduction section. 

The study can be accepted for publication after addressing some minor comments of mine.

-Please label the important peaks in the FT-IR figure.

-L159-161, please provide a reference.

-L161-163, reference, please.

-If possible please provide SEM analysis of the films to see the fibers distribution in more detail.

-The results are well discussed and the overall presentation is excellent. 

-Conclusion needs a little improvement in terms of future use of impact of the study on wood-industry.

Author Response

Dear Reviewer,

Thank you for your appreciation. We revised our manuscript according to your helpful comments.

In the current study, the researchers have proposed a new approach for the etherification of wood to impact thermoplasticity. The approach is quite interesting and will save both energy and resource on an industrial scale. The study is well presented. Especially the authors have successfully given all the background info and hypothesis in the introduction section.

The study can be accepted for publication after addressing some minor comments of mine.

Ans. Thank you for your positive comments.

-Please label the important peaks in the FT-IR figure.

Ans. We labeled important peaks in the FT-IR spectra in Figure 1 (Line 188 in the revised manuscript).

-L159-161, please provide a reference.

Ans. We added the following reference (Line 162 in the revised manuscript), [12] Norimoto, M.; Morooka, T.; Aoki, T.; Shiraishi, N.; Yamada, T.; Tanaka, F. Some physical properties of benzylated wood. Mokuzai Kenkyu Shiryou 1983, 17, 181–191. (in Japanese)

-L161-163, reference, please.

Ans. We added the following reference (Line 162 in the revised manuscript), [20] Shi, J.; Xing, D.; Li, J. FT-IR studies of the changes in wood chemistry from wood forming tissue under inclined treatment. Energy Procedia 2012, 16, 758-762.

Along with this, the numbers of subsequent references have been revised (Line 305 and 334 in the revised manuscript).

-If possible please provide SEM analysis of the films to see the fibers distribution in more detail.

Ans. Thank you for your fruitful comment. The preparation of various materials using the benzylated wood powder obtained in this study is the important issue. Actually, the structural observation of the obtained film and the characterization such as physical property analysis are now undertaken. Since there are a number of data regarding the characterization of the obtained material, we would like to summarize them in a next paper.

-The results are well discussed and the overall presentation is excellent.

Ans. Thank you for your positive comment.

-Conclusion needs a little improvement in terms of future use of impact of the study on wood-industry.

Ans. We added a sentence about the improvement in terms of future use of wood in Conclusions as follows:

“This benzylation method renders thermoplasticity to wood in a short time without heating, therefore it is expected to contribute to promoting the use of wood as an industrial material.” (Line 414 in the revised manuscript)

Reviewer 2 Report

The paper focuses on a novel procedure for a rapid benzylation of wood, performed at room temperature. The manuscript is well written, and the authors have performed sufficient experimental procedures/characterization.

I believe that the manuscript is suitable for publication, after some minor revisions:

Page 4, line 162: “(Figure 1, blue circle).” The authors should correct the “blue circle” by “blue line”.

Page 4, lines 165: “The intensities of these peaks decreased after the benzylation reaction (entry 1) because of the alkaline treatment. The acetyl groups were cleaved from hemicellulose during the treatment. In addition, the carbonyl groups in lignin were probably oxidized to carboxyl groups and then to carboxylates.”

I agree with the authors regarding the cleavage of the acetyl groups. However, when the carboxylation occurs, the C=O band remains in the FTIR spectra and will not be decreased as the authors stated. Please correct the information.

Page 7, line 259: “it is possible that some of the [(n-Bu)4P]OH was thermally decomposed at higher temperatures.”

Is it possible, that another plausible explanation for the low substitution degree at higher temperatures, is the inactivation of the BnBr by the [(n-Bu)4P]OH? Since this is a base, at higher temperatures, the BnBr can react with OH-, leading to the formation of BnOH, which will be no longer suitable to react with the wood.

Author Response

Dear Reviewer,

Thank you for your appreciation. We revised our manuscript according to your helpful comments.

The paper focuses on a novel procedure for a rapid benzylation of wood, performed at room temperature. The manuscript is well written, and the authors have performed sufficient experimental procedures/characterization.

I believe that the manuscript is suitable for publication, after some minor revisions:

Ans. Thank you for your positive comments.

Page 4, line 162: “(Figure 1, blue circle).” The authors should correct the “blue circle” by “blue line”.

Ans. It was a typo. Thank you for your advice. We corrected it to “blue line”. (Line 162 in the revised manuscript)

Page 4, lines 165: “The intensities of these peaks decreased after the benzylation reaction (entry 1) because of the alkaline treatment. The acetyl groups were cleaved from hemicellulose during the treatment. In addition, the carbonyl groups in lignin were probably oxidized to carboxyl groups and then to carboxylates.”

I agree with the authors regarding the cleavage of the acetyl groups. However, when the carboxylation occurs, the C=O band remains in the FTIR spectra and will not be decreased as the authors stated. Please correct the information.

Ans. You are right. The peak intensity of FT-IR does not decrease even if the carbonyl group is changed to a carboxyl group. The decrease in IR peak intensity derived from lignin is considered to be due to dissolving into the alkaline solvent, [(n-Bu)4P]OH, due to the oxidative decomposition of lignin. Therefore, we added the following sentence:

"Because of this oxidative decomposition, some amount of lignin dissolved in the alkaline solvent." (Line 170 in the revised manuscript)

Page 7, line 259: “it is possible that some of the [(n-Bu)4P]OH was thermally decomposed at higher temperatures.”

Is it possible, that another plausible explanation for the low substitution degree at higher temperatures, is the inactivation of the BnBr by the [(n-Bu)4P]OH? Since this is a base, at higher temperatures, the BnBr can react with OH-, leading to the formation of BnOH, which will be no longer suitable to react with the wood.

Ans. Thank you for your helpful advice. We added several sentences to that part as follows:

“Another possibility of low degree of substitution is the inactivation of BnBr by [(n-Bu)4P]OH. BnBr reacts with OH anion of the strong basic solvent to form BnOH, which does not react with wood. This inactivation of BnBr might have been promoted at higher temperature.” (Line 262 in the revised manuscript),
